# Research Advances on Hydrogel-Based Materials for Tissue Regeneration and Remineralization in Tooth

**DOI:** 10.3390/gels9030245

**Published:** 2023-03-20

**Authors:** Zhijun Zhang, Fei Bi, Weihua Guo

**Affiliations:** 1State Key Laboratory of Oral Diseases, West China Hospital of Stomatology, Sichuan University, Chengdu 610041, China; 2National Engineering Laboratory for Oral Regenerative Medicine, West China Hospital of Stomatology, Sichuan University, Chengdu 610041, China; 3Department of Pediatric Dentistry, West China School of Stomatology, Sichuan University, Chengdu 610041, China; 4Yunnan Key Laboratory of Stomatology, The Affiliated Hospital of Stomatology, School of Stomatology, Kunming Medical University, Kunming 650500, China

**Keywords:** scaffolds, pulp regeneration, periodontal regeneration, remineralization

## Abstract

Tissue regeneration and remineralization in teeth is a long-term and complex biological process, including the regeneration of pulp and periodontal tissue, and re-mineralization of dentin, cementum and enamel. Suitable materials are needed to provide cell scaffolds, drug carriers or mineralization in this environment. These materials need to regulate the unique odontogenesis process. Hydrogel-based materials are considered good scaffolds for pulp and periodontal tissue repair in the field of tissue engineering due to their inherent biocompatibility and biodegradability, slow release of drugs, simulation of extracellular matrix, and the ability to provide a mineralized template. The excellent properties of hydrogels make them particularly attractive in the research of tissue regeneration and remineralization in teeth. This paper introduces the latest progress of hydrogel-based materials in pulp and periodontal tissue regeneration and hard tissue mineralization and puts forward prospects for their future application. Overall, this review reveals the application of hydrogel-based materials in tissue regeneration and remineralization in teeth.

## 1. Introduction

Oral health is one of the important factors affecting physical health and quality of life [1]. Oral diseases are an increasingly serious public health problem in both developed and developing countries [2]. Common oral diseases include dental caries, pulp necrosis, periodontitis, tooth demineralization, etc. [3]. Regeneration of dental tissue and re-mineralization of teeth is important for oral health. Tissue engineering provides great hope for tissue regeneration and remineralization in teeth. The tooth is a complex organ with soft and hard tissues of different properties, such as enamel, cementum, dentin, periodontal membrane and pulp. The teeth have a complex root canal network and a sandwich structure of cementum-periodontal membrane-alveolar bone. Because the root canal system is limited by the unique anatomy of the long, narrow cavity, it is difficult to fill up the entire canal with preformed material. The unique physiological position and multilayer structure of periodontal tissue also put forward high requirements for materials. The re-mineralization of the hard tissue of teeth requires fibers as templates, Fortunately, with the rapid development of polymer materials, the demands for biosafety, fluidity and versatility of materials to meet requirements in dental tissue repair practices are promisingly fulfilled.

Hydrogel-based materials, as one of the scaffold materials in tissue engineering, have attracted more and more attention [4,5,6]. Hydrogel is a high polymer crosslinked network with three-dimensional stability formed by hydrophilic polymer and provides a porous hydrophilic microenvironment that facilitates the diffusion of oxygen and nutrients. Due to the strong hydrophilicity of hydrogel, its structural characteristics can remain stable for a long time after swelling in water. When hydrogels are cured in a specific environment, they can also maintain a certain volume and shape. Thermosensitive hydrogels [7] and photosensitive hydrogels [8] form fixed shapes when exposed to specific temperatures or wavelengths of light. Based on the physical and chemical properties of hydrogels, they are widely used in the research of tissue engineering regeneration, especially for teeth with complex anatomical structures. Biocompatible hydrogels include injectable hydrogels [9], tissue-related hydrogels [10], nanocomposite hydrogels [11] and intelligent hydrogels [12]. Functional hydrogels are sensitive to physiological and pathological signals and have a wide range of biomedical applications [13].

We used the PubMed website to search for research advances on hydrogel-based materials for tissue regeneration and remineralization from 2014 to 2023. This manuscript briefly reviews the application of hydrogels in the tissue engineering of teeth and periodontal tissues and the prospects for their further application in the field of tooth regeneration.

## 2. Application of Hydrogel in Pulp Tissue Regeneration

Regenerative dentistry has made considerable progress from pulp cap to pulp regeneration, to regenerate the pulp–dentin complex and restore its function that has been impaired by pulp injury and/or inflammation. Due to the unique anatomical limitations of tooth root canal structure, designing a suitable microenvironment to facilitate vascular/angiogenesis and innervation is a challenging task. Biomedical approaches to creating regenerative microenvironments are characterized by stent-based or stentless strategies. Stent-based strategies rely primarily on the use of biological materials to create a structural foundation that supports cells throughout tissue formation. Stent-free methods use cell sheets, spheres, or tissue chains as building blocks. Combining these two approaches, through the use of cells in combination with scaffolds, may represent an optimal solution to circumvent some of the major disadvantages of current approaches in pulp regeneration while cultivating their advantages [14]. As the root canal system of teeth is complex and irregular, hydrogel as a carrier has great advantages in the research related to pulp regeneration [15]. Hydrogels of various materials can be loaded with stem cells, bioactive peptides, antibacterial drugs, etc., combined with injection, microspheres or 3D printing for the relevant research of pulp regeneration (Figure 1).

### 2.1. Different Types of Hydrogels for Pulp Regeneration

Multiple biomaterial hydrogels have potential in the research of pulp regeneration (Table 1), such as chitosan (CS) [16,17], fibronectin (FN) [18] platelet-rich fibrin (PRFe) [19], hyaluronic acid (HA) [20], bioactive glass (BAG) [21], polyethylene glycol (PEG) [22], silk fibroin (SF) [23], synthetic clay [24], carboxymethyl cellulose-hydroxyapatite hybrid [25], cinnamaldehyde (CMA) [26] and alginate [27].

### 2.2. Effect of Hydrogel on Dental Pulp Stem Cells

Stem cells play an important role in tissue regeneration, as do dental pulp stem cells. Based on the excellent physical and chemical properties of hydrogels and their potential as cell scaffolds, there is abundant research on the proliferation and differentiation of dental pulp stem cells (DPSCs). RADA16-I is an ion-complementary self-assembled peptide with a regular folded secondary conformation and can be assembled into an ordered nanostructure. Liu et al. used a self-assembled peptide RAD/Dentonin with an A-based nanofiber network structure to attach to hydrogels (L-gel or D-gel) by heat-cooling technology. It has good biocompatibility and can promote the adhesion, proliferation, migration, odontogenic differentiation and mineralization of human dental pulp stem cells (hDPSCs) [28]. In addition, Puramatrix™, a self-contained peptide hydrogel, has been shown to maintain the viability of DPSCs and promote their odontogenic differentiation [29]. Based on in vitro studies of stem cells with self-loaded peptides, its role was further explored by Kerstin et al. using a mouse model. DPSCs were coated with self-assembled peptide nanofiber hydrogel and growth factors (GFs). Fibroblast growth factor alkaline, transforming growth factor β1 and vascular endothelial growth factor was added to the hydrogel using heparin as a carrier. Soft connective tissue with blood vessels formed under the skin of immunocompromised mice [30]. Magnetic nanoparticles were then applied in the process of pulp regeneration by aggregating cells into cellular spheres. The formation of multicellular spheres on hydrogels promoted osteogenic differentiation of DPSCs by regulating the mechanical/interfacial properties of the hydrogels [31]. Meanwhile, exosomes of DPSCs were also loaded into hydrogels. A hydroxypropyl chitin (HPCH)/chitin whisker (CW) thermosensitive hydrogel loaded with exosomes of hDPSCs can regenerate soft tissue similar to dental pulp in vivo [32].

### 2.3. Application of Hydrogel Microspheres in Pulp Regeneration

In addition to injectables, hydrogels can be prepared into microspheres in which cells or bioactive factors can be loaded for pulp regeneration. (Figure 2) Hydrogel microspheres have attracted wide attention as delivery carriers of stem cells and factors in dental pulp regeneration medicine due to their advantages such as injectability, rapid material transfer ability and 3D simulation of the primary environment. Atila et al. combined injectable hyaluronic acid hydrogel (HAH) with RG1-loaded chitosan microspheres (CSM) to trigger odontoblast differentiation of hDPSC and vascularization of dental pulp. At the gene expression level, hydrogel-cultured DPSCs showed odontogenic differentiation (COL1A1, ALP, OCN, Axin-2, DSPP and DMP1) and angiogenic differentiation (VEGFA, VEGFR2 and eNOS) [33]. Zhang et al. prepared Gd-alginate/laponite (RGD-Alg/Lap) hydrogel microspheres with an average size of 350~450 μm by the electrostatic microdroplet method. When loaded with hDPSCs and vascular endothelial growth factor (VEGF), they could conduct the slow release of VEGF and promote the differentiation of human dental pulp stem cells. Eventually, it regenerates pulp-like tissue rich in blood vessels [34]. Yang et al. used methyl acryloyl gelatin (GelMA) as raw material, prepared GelMA microspheres by electrostatic microdroplet method, and investigated the potential application of GelMA microspheres in root canal regeneration. hDPSCs can effectively adhere to, diffuse, proliferate and secrete extracellular matrix proteins in microspheres, and tend to occupy the outer layer. In addition, the cell-carrying GelMA microsphere system was able to tolerate cryopreservation and the cells functioned normally after thawing. After subcutaneous implantation in nude mice, the cell-carrying GelMA microsphere group produced more vascularized medullary tissue with appropriate degradation rates than the cell-carrying bulk GelMA group [35].

Natural dental pulp contains abundant growth factors. The hydrogel prepared using a natural dental pulp decellularity matrix can fully simulate the microenvironment of dental pulp regeneration [36]. Zheng et al. conducted a further study on the role of the hydrogel microspheres, using the hydrogel microspheres to support the extracellular matrix of the dental pulp, release bioactive factors and simulated the specific three-dimensional (3D) microenvironment of dental pulp, which has good plasticity and biocompatibility and can promote the proliferation and differentiation of DPSCs [37].

### 2.4. Effect of Hydrogels on 3D Pulp Regeneration

Though hydrogels have been proven to influence the proliferation and differentiation of hDPSCs in vitro, there is still a lack of verification of their effect on cells in vivo. Pulp tissue is a complex connective tissue, and it needs a suitable regeneration scaffold to balance fluidity which allows DPSCs to differentiate from pulp tissue in a relatively stable situation. The 3D structural construction of hydrogels for DPSCs cultivation should be adjusted according to the root canal’s morphological characteristics, such as shape and size. DPSCs were cultured in the space of a hydrogel-based container, and a 3D hollow cylindrically shaped cell cultivation system with a length of 7 mm and a diameter of 3 mm was obtained. After 20 days of culture in vitro, the survival rate of culture cells remained above 85% [38]. Itoh et al. constructed a 3D DPSCs structure with heat-responsive hydrogel, which was filled into the root canal of human teeth and formed endodontic tissue with rich blood vessels after 6 weeks implanted subcutaneously into immunodeficient mice [39]. Hydrogels are also used in 3D printing-related research because of their plasticity, which properly serves as a component of the printing ink. The use of 3D-printed Alg–Gel scaffolds can promote the proliferation of DPSCs more than Alg–Gel scaffolds [40].

### 2.5. Regeneration of Vascular Nerves in Dental Pulp by Hydrogel

Pulp tissue is rich in blood vessels and nerves, which is an important structure for pain perception and tooth localization. Hydrogel can combine DPSCs with other kinds of cells for pulp regeneration, regenerating pulp-like tissues rich in blood vessels and nerves, and repairing the function of natural dental pulp. Biomimetic acellular peptide hydrogel can simulate the morphology, structure and physiological properties of natural pulp, and perform good biointegration and soft tissue regeneration in the Beagle canine pulp loss model [41]. hDPSCs and human umbilical vein endothelial cells (HUVECs) were mixed in 5% GelMAand injected into tooth root segments (RSs). These hydrogels loaded with a mixed cell suspension can attach to the dentin surface of RSs and infiltrate into the dentin tubules [42]. A cell-laden microfiber based on GelMA is used to carry seed cells and has achieved a good regeneration effect [43]. Cranial neural crest-derived cells (CNCLCs) obtained from induced pluripotent stem cells (iPSCs) were attached to the hydrogel to form a pulp-like tissue, such as harboring odontoblast-like cells, a dentin-like layer, and vast neovascularization [44].

Zhang et al. prepared a functional GelMA microsphere system combining human platelet lysate (PL) with nano-clay Laponite by electrostatic microdroplet method. The results showed that GelMA/PL/Laponite microspheres significantly promoted the diffusion, proliferation and odontogenic differentiation of encapsulated hDPSCs compared with GelMA microspheres alone. In addition, they promote the formation of microvessels and the regeneration of medullary tissue in the body. This hybrid microsphere system has great potential to promote the formation of microvessels in regenerated dental pulp and other tissues [45]. In order to improve the cell survival rate and pulp tissue regeneration ability of stem cells from human exfoliated deciduous teeth (SHED), Han et al. used lentiviral short hairpin RNA to inhibit proline hydroxylase domain protein 2 (PHD2) to stabilize hypoxia-inducing factor 1α (HIF-1α) and pretreated it to hypoxia state. Hif-1α-stabilized SHED was encapsulated in PuraMatrix hydrogel and injected into the root canal of human dental fragments and implanted into the subcutaneous space of immunodeficient mice. Twenty-eight days later, enhanced pulp tissue formation and significantly increased vascularization levels were observed, which may be related to SHED endothelial differentiation and host vascular recruitment [46].

### 2.6. Antibacterial Effect of Hydrogels on Pulp Regeneration

The main reason for the loss of natural pulp is the invasion of bacteria into the root canal system; bacteria that remain in the root canal lumen during regeneration may contaminate the implanted biological material. Therefore, the antibacterial effect of bioactive materials is also a particularly important link in the process of pulp regeneration. A variety of biomaterials have been shown to reduce inflammation in the root canal and thus facilitate pulp regeneration, such as silver-doped bioactive glass/chitosan hydrogel [47] and mesoporous silica-loaded metronidazole composite hydrogel [48]. An innovative fibrin hydrogel incorporating clindamycin (CLIN)-loaded poly (d, l) lactic acid (PLA) nanoparticles (NPs) can improve the antibacterial properties of hydrogel and promote pulp regeneration without affecting cell activity and function [49]. An injectable photo crosslinked GelMA was successfully fabricated with ciprofloxacin (CIP) for oral infection ablation. To this end, CIP or its β-cyclodextrin (β-CD)-inclusion complex (CIP/β-Cd-IC) was incorporated into polymer electrospun fibers, which were cut into short nanofibers and then embedded in GelMA to obtain injectable hybrid antibacterial hydrogels. Due to the enhanced CIP solubility of beta–CD–IC and the tunable degradation properties of GelMA, hydrogels promote topical, sustained, and effective doses of cell-friendly antibiotics that have inhibitory effects on the growth of Gram-positive Enterococcus faecalis [50]. Ribeiro et al. used chloro-bound (CHX)-loaded kaolin aluminosilicate nanotubes (HNTS) modified GelMA hydrogel to provide sustained CHX release for ablation of infection while showing good biocompatibility. In a rat subcutaneous model, (CHX) -loaded nanotube-modified GelMA showed minimal local inflammatory response. Moreover, the addition of ChX-supported nanotubes increased the swelling rate and decreased the degradation rate of the hydrogel. Importantly, ChX-loaded nanotubes inhibited bacterial growth with minimal cytotoxicity [51].

Hydrogels and their derivatives can promote the regeneration of dental pulp tissue by promoting the proliferation, migration and differentiation of DPSCs, mixing various cells or carrying different bioactive factors to provide a suitable microenvironment for dental pulp regeneration.

## 3. Application of Hydrogel in Periodontal Tissue Regeneration

Periodontitis occurs when the host immune system and plaque microorganisms interact simultaneously. Periodontitis is considered to be a major pathogenic cause of tooth loss. The main reason is the loss of alveolar bone when periodontal inflammation occurs. The resorption of alveolar bone is irreversible. The research focus of periodontal regeneration in recent years is mainly on anti-inflammatory and periodontal tissue regeneration. Similar to pulp tissue regeneration, hydrogels can also carry drugs, microspheres and cells in the process of periodontal tissue regeneration. Due to the unique physiological structure of periodontal tissue and the pathogenesis of periodontitis, the application of hydrogel in periodontal tissue is also quite different from other tissues, which is mainly reflected in the following aspects (Figure 3).

### 3.1. Hydrogels Loaded with Drugs or Bioactive Factors Promote Periodontal Regeneration

Hydrogels containing drugs with both anti-inflammatory and tissue regeneration properties can effectively promote periodontal regeneration in periodontitis, such as BMP-2/VEGF [52]; bone morphogenetic proteins (BMP) [53]; matrix metalloproteinase 8 (MMP-8) [54]; elf-assembling peptide (SAP) [55]; bone morphogenetic protein-7 (BMP-7) [56]; insulin-like growth factor-I [57]; recombinant human beta-nerve growth factor (rh beta-NGF) [58]; dexamethasone [59]; interleukin-1 receptor antagonist (IL-1ra) [60]; kinase 3 beta inhibitor (BIO) [61]; minocycline and zinc oxide nanoparticles (ZnO NPs) [62]; chlorhexidine (CHX) [63]; interleukin (IL)-4/stromal cell-derived factor (SDF)-1α [64]; chlorhexidine [65]; naringin [66]; metronidazole (MD) [67]; ornidazole [56]; triclosan (TCS)/flurbiprofen (FLB) [68] and moxifloxacin hydrochloride (Mox)/clove essential oil (CEO) [69] (Table 2).

### 3.2. Hydrogels Carry Cells or Cell-Related Products for Periodontal Regeneration

Exosomes derived from odontogenic stem cells can promote periodontal tissue regeneration by regulating macrophage phenotype. The injection of exosomes into periodontal tissue using hydrogels as carriers provide a feasible scheme for periodontal tissue regeneration. DPSC–Exo–chitosan hydrogel (DPSC–Exo/CS) can accelerate the healing of alveolar bone and periodontal epithelium in periodontitis mice. When dental pulp stem cell-derived exosomes were loaded into chitosan hydrogel, the regeneration of alveolar bone and periodontal fibers was accelerated in mice by injection [70].

### 3.3. Effect of Temperature–Sensitive/Photocured Hydrogel in Periodontal Regeneration

Due to the structural particularity of periodontal tissue, it is difficult for formed biomaterials to adapt to the morphology of periodontal tissue. However, temperature-sensitive hydrogels are a suitable material. When injected into the body, the constant temperature of the organism will make the gel solidify, which is conducive to preserving the cells and bioactive factors carried by the hydrogel in the periodontal region. CS/β-sodium glycerophosphate (β-GP)/gelatin hydrogels loaded with aspirin/erythropoietin (EPO) can simultaneously achieve pharmacological effects of anti-inflammatory and periodontal tissue regeneration through continuous release of aspirin and EPO [71]. Thermosensitive hydrogels play an important role regarding in situ gel administration. Heat-sensitive carriers with gelation temperatures in the range of 30–36 °C can be easily injected in liquid form and converted into gel after injection. By mixing PX with polyacrylic acid (PAA) and adjusting the gel temperature of poloxamer 407 (PX), a new type of near-thermal gel was prepared. The mixture behaves as a low-viscosity liquid at room temperature. The shear thinning behavior makes it injectable and rapidly gelates at body temperature [72]. Of course, photocured hydrogels have the same advantages in periodontal regeneration for they own excellent fluidity for injection and can after be solidified by light with a specific wavelength. It has been proved that 2-Methylimidazole zinc salt (ZIF-8), an injectable photopolymerized ZIF-8/GelMAcomposite hydrogel (Gelma-Z), can reduce inflammation and promote periodontal tissue regeneration when injected into periodontal defects of rats [73].

### 3.4. Application of Hydrogel in Periodontal Structure Reconstruction

Natural periodontal tissue includes three layers of cementum, periodontal membrane and alveolar bone. There are significant differences in the cells, morphology and structure of the three-layer tissues. Sowmya et al. used hydrogels to simulate the naturally layered structure of periodontal tissue. Three layers of different materials were used as scaffolds to achieve complete and synchronous regeneration of hard tissue (cementum and alveolar bone) and soft tissue (periodontal ligament (PDL) in rabbit periodontal defects. The cementum layer was simulated by chitin–polylactic acid–glycolic acid (PLGA)/nano bioglass ceramic (nBGC)/cementin 1, -PLGA/fibroblast growth factor 2 mimics periodontal membrane, and chitin-PLGA/nBGC/platelet-rich plasma-derived growth factor mimics alveolar bone layer [74]. Non-chemically measured wollastonite (nCSi) scaffolds and GelMA methacrylate/silanated hydroxypropyl methylcellulose (Gelma/Si-HPMC) hydrogel membranes were prepared by digital optical processing (DLP) and optical crosslinking hydrogel injection technology, respectively, which can regenerate periodontal tissue in periodontal defects of Beagle dogs [75].

In addition to simulating the multilayer structure of the periodontal membrane tissue, 3D printing technology is also applied in the study of periodontal tissue regeneration because of the high controllability of the scaffold structure. A bio print-based approach was constructed to generate nanometer-sized 3D cell-supported hydrogel arrays with ECM component gradients by controlling the volume ratios of two hydrogels, GelMA and PEGdimethacrylate [76]. Periodontal ligament stem cells (PDLSCs) were encapsulated in injectable photo crosslinked composite hydrogels consisting of GelMAand polyethylene glycol dimethacrylate (PEGDA). Pdlsc-loaded GelMA/PEGDA hydrogels with different compositions were prepared by controlling the volume ratio of GelMA to PEGDA on a 3D bioprinting platform. The optimized composition of PDLSC-loaded hydrogels resulted in the formation of robust new bone in the defects compared to the control group [77]. There have been studies applying GelMA to 3D print Bionic periodontium patches (BPPs) for functional periodontal regeneration. In Beagle dogs, ortho-transplantation of the mandible demonstrated effective periodontal reconstruction, producing a “sandwich structure” that closely resembled natural periodontal tissue and regenerated correctly oriented fibers [78].

### 3.5. Hydrogels Used for Periodontal Regeneration Associated with Systemic Diseases

Both periodontitis and diabetes are chronic diseases with high prevalence, and they are closely related. On the one hand, diabetes is a risk factor for periodontitis. On the other hand, periodontitis has a negative effect on the blood sugar control of diabetes. The correlation between the two has been a hot topic in recent years. Diabetes mellitus (DM) aggravates periodontitis and leads to accelerated resorption of periodontal bone. Impaired bone healing due to excessive production of reactive oxygen species (ROS) caused by disrupted glucose metabolism in diabetes makes diabetic periodontal bone regeneration a major challenge. A mesoporous silica nanoparticle (MSN) combined with PDLLA (poly (DL-lactic acid)-PEG-PDLLA (PPP) thermosensitive hydrogel was designed to progressively release load to simulate the mesenchymal stem cell “recruitment-osteogenic” cascade for diabetic periodontal bone regeneration. In vivo results show that it can effectively recruit rBMSCs to periodontal defects and significantly promote periodontal bone regeneration in animal models of type 2 diabetes [79].

Periodontitis is a kind of local periodontal microecological disorder caused by the imbalance between oral flora and host immune response. Macrophages are an important part of the host immune response. As the main antigen-presenting cells, macrophages play an important role and function in the occurrence, development and outcome of periodontitis. Porphyromonas gingivium (p.g.) is a key group of bacteria in the pathogenesis of periodontitis. A nanoparticle formulation (MZ@PNM)-enveloping hydrogel (MZ@PNM@GCP) based on penetrating macrophages can target the toll-like receptor complex 2/1 (TLR2/1) on the membrane through its macrophages. Then, cationic nanoparticles destroy the structural integrity of bacteria and release the antibacterial drug metronidazole (MZ) in the cell to directly kill P.g. MZ@PNM@GCP shows strong efficacy in treating periodontitis, restoring local immune function, killing pathogenic bacteria, and has good biocompatibility, which has been confirmed in vivo and in vitro experiments [80].

### 3.6. Clinical Study on the Application of Hydrogel to Periodontal Regeneration

Clinical studies on hydrogels in periodontal regeneration have been underway. Human fibroblast growth factor 2 (rhFGF-2) biological hydrogels were applied in hyaluronic acid (HA) carrier to treat periodontal defects. A total of 30 adult patients were evaluated, and the clinical parameters of periodontal wound healing after 1 year of treatment were significantly improved [81].

Periodontal tissue regeneration is the regeneration and functional reconstruction of alveolar bone, cementum and the periodontal membrane. For a long time, much research has been carried out to find an ideal method to repair the damaged periodontal tissue as well as possible. A large number of studies have confirmed that hydrogel can be used as a potential material for periodontal tissue regeneration.

## 4. Application of Hydrogel in Hard Tissue Regeneration of Teeth

In recent years, with the increase of the aging population and changes in diet structure, tooth and bone problems have become more prominent, such as tooth decay and osteoporosis. The damage to these hard tissues is often irreversible, and it is often necessary to implant alternative biological materials to repair the defects clinically. Caries is a kind of chronic progressive irreversible damage to the hard tissue of teeth. Remineralization after caries is very important for the repair of the hard tissue of teeth. Biomineralization is an important link in the formation of hard tissues during the development of organisms, including the shell of molluscs and bones and teeth of vertebrates [82]. Cells, proteins, and organic and inorganic substances in organisms all play an important role in the formation of inorganic minerals [83]. The substantial process of biomineralization is the nucleation and assembly process of inorganic substances in vivo under the control of organic matter [84]. It is an important biological strategy for organisms to regulate mineral deposition and enhance tissue function through minerals (Figure 4).

### 4.1. Promotion of Mineralization of Odontogenic Stem Cells by Hydrogel

Stem cells play an important role in the process of biomineralization. It is an important biological strategy for hard tissue regeneration of teeth to regulate mineral deposition and enhance tissue function by promoting osteogenic differentiation and odontogenic differentiation of cells. The study of hydrogel on the mineralization of odontogenic cells is the basis of hard tissue regeneration of teeth (Table 3).

During tooth development, ectoderm-derived ameloblasts produce enamel by synthesizing a complex mixture of proteins that control cell-matrix interactions and the habit of hydroxyapatite crystals. Huang et al. investigated the effect of artificial bioactive nanostructures on ameloblasts in a hydrogel culture system using an in vitro cell and organ culture system. Using a branched peptide amphiphilic molecule containing the Arg–Gly–Asp peptide motif, or “RGD” (abbreviation BRGD–PA), Ameloblasts (line LS8) and primary enamel organ epithelium (EOE) cells were cultured in PA hydrogel and injected into the enamel organ epithelium of mouse embryonic incisors. It has been shown that BRGD–PA nanofibers containing enamel proteins are involved in integrin-mediated cell-matrix binding and transmit guiding signals for enamel formation [96].

As a mature scaffold material for tissue regeneration, Bhatnagar et al. confirmed through experiments that hydrogels can increase the mineralization ability of DPSCs. When the DPSCs were inoculated in hard (~8 kPa) and soft (~0.15 kPa) gelatin hydrogels for 35 days, odontogenic differentiation markers such as OCN, ALP and DSPP were up-regulated. Scanning electron microscopy (SEM) and alizarin red (alizarin red) staining showed the DPSCs deposited a layer of hydroxyapatite bimineralized flakes on the surface and inside of the hydrogel, demonstrating the potential of enzyme-crosslinked gelatin hydrogels as scaffolds for dentin regeneration [97].

### 4.2. Application of Hydrogel in Enamel Regeneration

Caries is a chronic, progressive and destructive disease that occurs in the oral cavity and is harmful to human life and health. The pathological change is the decomposition of organic matter in tooth enamel and demineralization of inorganic matter, which is a continuous dynamic chemical reaction. The demineralization of tooth enamel can easily occur in orthodontics [98], and the re-mineralization of teeth can effectively slow down the sensitive symptoms of enamel and prevent early caries. Hydrogels and hydrogel-related products create a promising microenvironment for enamel re-mineralization, showing remarkable effects (Table 4).

#### 4.2.1. Hydrogels Carry Cellular or Bioactive Factors to Promote Enamel Re-Mineralization

In addition to the effect on the differentiation of odontoblast-related cells, a variety of hydrogels on the regeneration of enamel have been confirmed. As a carrier for cells or growth factors, injectable hydrogels offer great potential for enamel regeneration. It is important to note that the development of injectable hydrogels with the appropriate structure and properties has been a challenging task, leaving much to be desired in terms of cytocompatibility, antibacterial characteristics, self-healing properties, and the ability to support dental stem cell function.

Mohabatpour et al. used a novel self-crosslinking hydrogel composed of oxyalginate and carboxymethyl chitosan to study enamel re-mineralization, which showed significant antibacterial properties against two caries-causing bacteria, Streptococcus mutans and Streptococcus brown. HAT-7 cells coated in hydrogel showed alkaline phosphatase production and mineral deposition and maintained their rounded morphology after 14 days in vitro. This study provides evidence that oxyalginate—carboxymethyl chitosan hydrogels can be used as injection cell carriers for tissue engineering applications in tooth enamel [109]. Ruan et al. developed a novel amyloidin-containing chitosan hydrogel for enamel reconstruction that stabilizes calcium–phosphorus clusters and guides their arrangement into linear chains through supramolecular assembly of amyloidins. The newly grown enamel layer forms a dense interface with the natural enamel layer. Compared with the computed enamel layer, the hardness and elastic modulus of the enamel layer are improved. Chitosan hydrogels can also protect against secondary caries due to their ph responsiveness and antibacterial properties [113]. Matrix metalloproteinase-20 (MMP-20) is encapsulated in CS–AMEL hydrogel to enhance the mechanical system function and promote the regeneration of biomimetic tooth enamel. The results showed that MMP-20-CS-AMEL hydrogel significantly increased the modulus and hardness of the repaired enamel (1.8 times and 2.4 times, respectively) [103].

Biomimetic synthesis of artificial enamel is a promising strategy for preventing and repairing enamel defects. Cao et al. designed a hydrogel biomimetic mineralization model for the regeneration of enamel-like mineralized structures with prismatic structures. By scanning electron microscopy, X-ray diffraction, Fourier transforms infrared spectroscopy and nanoindentation hardness tests, the mineralized structure of the model on the surface of etched enamel in the presence of 500 ppm fluoride was analyzed. The resulting tissue has an enamel prismatic layer containing well-defined hexagonal hydroxyapatite crystals. The elastic modulus and nano hardness of regenerated enamel prismatic structure are similar to those of natural enamel. Therefore, using this hydrogel biomimetic mineralization model to regenerate tooth enamel is a promising method for the treatment of tooth enamel loss [106]. Agarose hydrogels have also been used to study dentin re-mineralization in vivo [114].

#### 4.2.2. Hydrogels Carry Biomimetic Polypeptide to Promote Enamel Re-Mineralization

The key scientific issue in the process of raw mineralization is the regulatory effect of organic substrates on the deposition of inorganic ions and nucleation of crystals. In addition to collagen, some non-collagen proteins in the organic substrates of teeth can induce the mineralization process. There are a wide variety of common non-collagen proteins in teeth, including dentin phosphoprotein (DSPP), dentin matrix protein (DMP1), bone morphogenetic protein 2 (BMP-2), osteocalcin, and osteopontin. Odontogenic ameloblast-associated protein (ODAM) is mainly expressed in the mature stage of enamel formation and has strong interaction with AMTN. AMTN is involved in enamel mineralization, but the effect of ODAM on mineralization has not been studied. Ikeda et al. determined that ODAM hydrogels were able to induce HA mineralization in modified simulated body fluids (SBF) and in vitro collagen substrates [110]. Human enamel sections were demineralized with 37% phosphoric acid for 1 min, covered with 2 mm thick EMD-calcium chloride (CaCl2) agarose hydrogel and added with 2 mm thick ion-free agarose hydrogel to promote the formation of prismatic crystals in vitro [108]. Recently, recombinant full-length amelelogenin protein was combined with fluoride in hydrogels. Amelelogenin (rP172) containing calcium, phosphate and fluoride showed a significant re-mineralization effect in early enamel caries models in vitro. After releasing amylogenin from hydrogels, The surface microhardness of the re-mineralized enamel was significantly increased, and no cytotoxicity was observed when the periodontal ligament cells were cultured with mineralized hydrogel [111]. Lu et al. used a novel biomimetic hydrogel composite containing QP5 peptide and bioactive glass (BG) to simulate the enamel process, where QP5 promotes enamel re-mineralization by guiding calcium and phosphorus ions provided by BG. In addition, BG can regulate the mineralized microenvironment to be alkaline, simulating the pH regulation of ameloblasts during enamel maturation. BG composites have better biosafety, better enamel binding ability, ion release ability and pH buffering ability. Enamel NCLs treated with BG l composite showed a higher reduction in lesion depth and mineral loss both in vitro and in vivo [100].

At present, 3D printing technology provides great possibilities for tissue engineering regeneration. The bio-ink that can be used for scaffold printing is limited, especially in dental tissue engineering. There are studies using a new bio-ink made of carboxymethyl chitosan (CMC) and alginate (Alg) for the bio-printing scaffold of tooth enamel tissue regeneration. Because CMC has good antibacterial ability and cell interaction performance. Alg was used to improve its printability and mechanical properties and regulate the degradation rate. al4%–CMC2% and 2% al4%–CMC showed high potential to promote the differentiation of ameloblasts, calcium and phosphorus deposition and matrix mineralization [115].

### 4.3. Application of Hydrogel in Dentin Regeneration

Dentin is the largest hard tissue in teeth, and its damage is irreversible. Research on dentin regeneration using hydrogel as the carrier has also made some progress (Table 5).

With the development of new materials, novel technologies and in-depth study of the biological behavior of pulp, more and more dentists give priority to preserving living pulp in endodontic treatment practices, especially for deep caries or reversible pulpitis. Pulp histopathological studies showed that pulp exposure did not necessarily cause irreversible changes in the pulp. Especially when deep caries or reversible pulpitis were clinically diagnosed, the pulp could still return to a normal state after removing the irritation of the affected tooth. At the same time, the development of novel pulp-capping materials has greatly improved the success rate of direct pulp capping. Mineral trioxide aggregate (MTA), bioceramics material iRoot BP and bioactive dentin replacement material Biodentin have been proven to have good biocompatibility, sealing performance, antibacterial activity and bio-induction properties.

Inducing dentin formation on exposed pulp is a major challenge in the study of dentin-pulp complex regeneration. Naoki et al., respectively, used collagen sponge and collagen hydrogel with fibroblast growth factor 2 (FGF2) on dentin formation in exposed rat molars. The controlled release of FGF2 in gelatin hydrogel induced the formation of dentin-like particles with dentin defects above the exposed pulp [122,123].

Clinically, pulp covering materials are commonly used to form restorative dentin, protect pulp tissue, and prevent deep caries, accidental exposure of pulp or local pulp cutting. Traditional pith cap materials used clinically include calcium hydroxide and trioxide mineral aggregates. Using panax notoginseng saponin R1 (Gel-MA/NGR1) supported methacrylic acid functionalized gelatin as raw material, Wang et al. prepared injectable colloids that induced restorative dentin formation. Physical and chemical properties test results showed that the prepared hydrogels had a suitably connected porous microstructure, the pore size was 10.5 microns, the mechanical properties were good, the modulus was 50~60 kPa, and it was conducive to cell adhesion and proliferation. Hydrogels maintain hydrophilicity and continuous drug release, promote the odontogenic differentiation of mouse dental papilla cells, and have a strong ability to promote the formation of restorative dentin, which is a potential pulp-covering material [119].

Ahmed et al. used a novel injectable dentin matrix hydrogel (TDMH), Biodentine, and MTA to perform pulp cap surgery on traumatized permanent posterior teeth, and evaluated the dentine Bridges formed under the pulp cap material using CBCT imaging. A total of 45 patients with accidental traumatic pulp exposure were selected. During follow-up, all patients had no symptoms, clinical signs and symptoms, and no pathological signs on imaging. However, CT results showed that the material measured in the TDMH group had different degrees of influence on the dentine bridge formation, and its radiation density and thickness were significantly better than that of Biodentine and MTA [124]. The dentin matrix molecule (DMM) microparticle hydrogel prepared by Cunha et al. showed good pulp cells and inflammatory response in rat tooth tissue after being printed by digital light projection 3D printer and significantly increased the deposition of dentin [125].

### 4.4. Application of Hydrogel in Hard Tissue Disease of Teeth

Fluoride preparations are promising for the repair and prevention of caries, but their interaction with enamel is often hindered and diminished due to the dynamic moist environment in the mouth, which affects the effectiveness of fluoride delivery and limits the success of treatment. Using tannic acid (TA), silk fibroin (SF) and sodium fluoride (NaF) self-assembly as raw materials, Zhen et al. developed a wet bonding fluorination system based on mussel (TS@NaF). TS@NaF exhibits remarkable biostability and biocompatibility, has reliable wet adhesion, locally releases fluoride ions (F-), and induces significant deposition of calcium fluoride (CaF2) on enamel in vitro. In addition, TS@NaF has an anti-caries effect in vitro and induces a significant increase in enamel mineral density [126]. Wang et al. prepared a mineralized film composed of stable amorphous calcium phosphate (PAsp–ACP) nanoparticles using hydroxypropyl methyl cellulose (HPMC) and polyaspartic acid. HPMC contains multiple hydroxyl groups and is a film-forming material that can be dried to form dry films. In a humid environment, the film gradually turns into a gel. The results show that HPMC’s hydroxyl and methoxy groups can assist the stability of PAsp–ACP nanoparticles and maintain their biomimetic mineralization activity. The results further showed that the biostimulation mineralized membrane induced early mineralization of demineralized dentin after 24 h, and the entire demineralized dentin (3–4 µm) mineralization increased after 72 to 96 h [118].

Dentin hypersensitivity (DH) is a common dental symptom, which is caused by the exposure of nerve endings to the oral environment due to the denudation of dentin tubules, and the acute transient sharp pain caused by external stimuli such as temperature, machinery, wind, chemistry, etc. How to treat the pain caused by DH has become an urgent clinical problem. Occluding exposed dentin tubules by biomimetic mineralization and reproducing enamel-like tissue on the dentin surface is the first choice for long-term treatment. Currently, a variety of materials and methods have been used to treat DH, such as fluoride, KNO_3_, SnF_2_, oxalates, adhesives, bioactive glass, iontophoresis, etc. Varoni et al. proposed a new type of potassium oxalate-based hydrogel for the treatment of dentin allergy. Potassium oxalate hydrogel significantly reduced the permeability of dentin and blocked most of the occludable dentin tubules through crystallization precipitation, forming mineralized layers. Potassium oxalate-based hydrogels show better performance than commercially available products and artificial saliva and appear to be promising candidates for treating dentin allergies [127]. Ling et al. established a biomimetic mineralization model consisting of oligopeptide-stimulated dentin matrix protein 1 (DMP-1), trioxide mineral aggregates (MTA), and agar-hydrogel biomimetic mineralization model (AHBMM). DMP-1@MTA@AHBMM was used to block dentin tubules on the demineralized dentin surface. The enamel-like tissue is regenerated on the dentin surface, which contains fluorinated hydroxyapatite crystals. The microhardness of regenerated enamel was higher than that of demineralized dentin [128].

Hard tissue regeneration is the key and difficult point in tooth regeneration. Hydrogel can promote the mineralization of hard tissue through its own structural properties or by carrying a variety of bioactive factors. It is promising to apply hydrogel in clinical practice and to prospect for further development.

## 5. Conclusions and Future Perspectives

Hydrogels are widely used in tissue engineering research. This paper reviews the application of hydrogels in the regeneration of pulp, periodontal and hard tissue of teeth.

In terms of dental pulp regeneration, hydrogels are superior to many other biological materials due to their excellent physical and chemical properties, which can match highly personalized dental pulp cavity structures. Hydrogel has good biocompatibility with a variety of cells. It can be used as a medium to combine more than two kinds of cells to regenerate pulp tissue rich in nerves and blood vessels. Hydrogel microspheres can be used as a good sustained-release carrier to release drugs over different time periods.

In terms of periodontal tissue regeneration, hydrogels display excellent properties of biocompatibility, proper water retention and controlled release, which support cell differentiation, cell interactions, reducing inflammatory responses, and regulating immune reaction during periodontal regeneration Thermosensitive/photosensitive hydrogels can form a better connection with the original periodontal structure in a short time due to their characteristic of being flowable to adapt to the defect region and being solidified under proper temperature in vivo, which is conducive to the retention of implant materials in the lesion so that the cells, drugs and bioactive factors carried in the hydrogels can play a better role. Combined with the printability of hydrogels, scaffolds similar to natural periodontal tissue can be reproduced, which lays a good foundation for the application of hydrogels in periodontal tissue regeneration.

In the aspect of hard tissue regeneration, the related research is not as extensive and in-depth as pulp and periodontal tissue. Based on the principle of biomineralization, the combination of hydrogel and biomimetic polypeptide has brought the dawn of the regeneration of tooth-hard tissue. Though hydrogels have been shown to be effective in preventing caries, dentin hypersensitivity, and pulp capping, the challenge of being applied to translational research and clinical practices remains great.

The ideal hydrogel scaffold material should not only have excellent biological properties, simulate extracellular matrix, participate in the release of signal molecules, and regulate the behavior of stem cells, but also have excellent physical and chemical properties, including appropriate strength, appropriate porosity, degradation rate matching with tissues and other characteristics. Although many types of hydrogels for tooth regeneration have been developed, several key challenges remain: (1) Improve the biocompatibility of hydrogel materials so that the differentiation of odontogenic stem cells can be better induced. (2) Mechanical properties should be adjusted to better suit the hardness requirements of different tooth structures. (3) More convenient preservation methods need to be researched. (4) Clinical research still has great potential. (5) Cementum is an important part of periodontal tissue, but research on cementum regeneration is lacking. Although the results of most studies are promising, substantial clinical studies are still needed to determine the efficiency of the systems being prepared.

## Figures and Tables

**Figure 1 gels-09-00245-f001:**
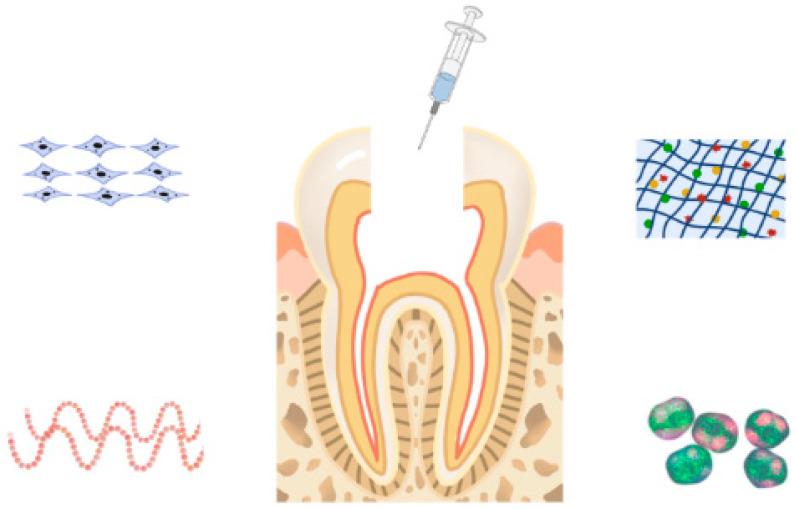
Application of hydrogel in pulp tissue regeneration.

**Figure 2 gels-09-00245-f002:**
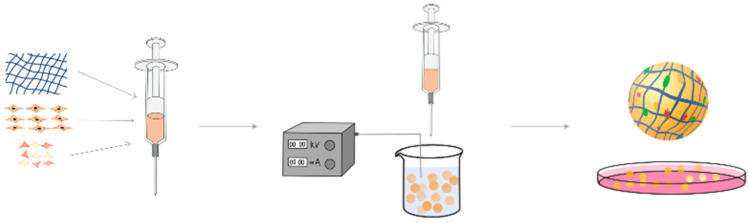
Application of hydrogel microspheres in pulp regeneration.

**Figure 3 gels-09-00245-f003:**
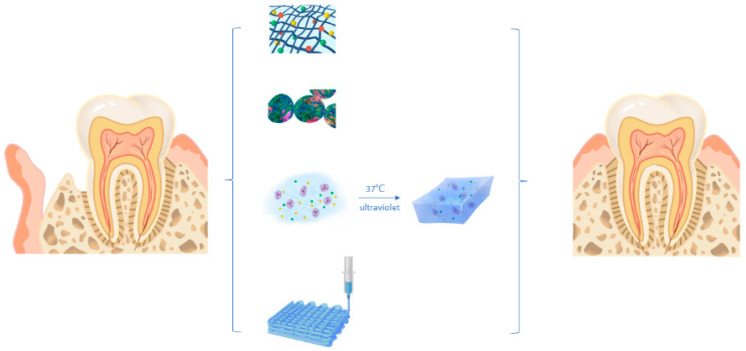
Application of hydrogel in periodontal tissue regeneration.

**Figure 4 gels-09-00245-f004:**
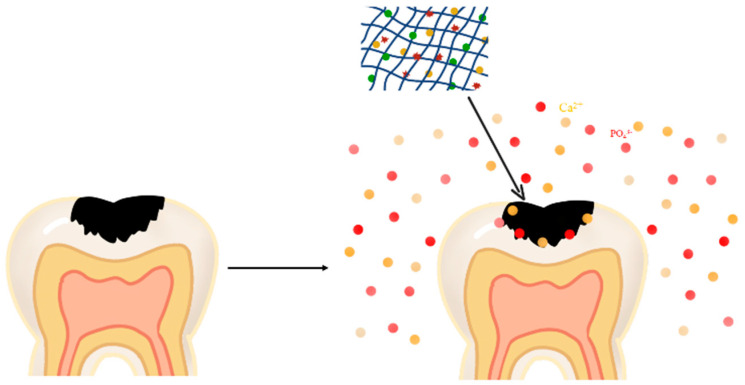
Application of hydrogel in hard tissue regeneration of teeth.

**Table 1 gels-09-00245-t001:** Different types of hydrogels for dental pulp regeneration.

Hydrogel	Cells	Achievement	Year	Ref.
chitosan	DPSCs	Chitosan is beneficial to promote the regeneration of human pulp tissue due to its antibacterial effect	2019	[16,17]
	Stemcell from the apical papilla (SCAPs)	Chitosan hydrogel can improve the effect of pulp regeneration through cell homing method	2021	
fibronectin	DPSCs	Collagen/gelatin hydrogel supplemented with 10 μg/mL FN showed strong bioactivity and chemotaxis on hDPSCs cultured in vitro	2021	[18]
platelet-rich fibrin	SCAPs	The novel PrFe-loaded ChitMA/ColMA hydrogel is injectable, cytocompatible, chemically attractive and bioactive to promote odontogenic differentiation	2022	[19]
hyaluronic acid	human bone marrow mesenchymal stem cells (hBMSCs)	Hyaluronic acid can maintain cell viability and proliferation, and promote osteogenic/dentin differentiation of hBMSCs	2022	[20]
bioactive glass	DPSCs	Bioactive glass microparticles enhance the osteogenic differentiation of DPSCs	2020	[21]
polyethylene glycol	DPSCs	Hydrogels promote the aggregation of DPSCs and their odontogenic differentiation	2015	[22]
silk fibroin	DPSCs	Pulp stem cells cultured with silk fibroin combined with gel showed good growth, proliferation and osteogenic differentiation ability	2022	[23]
synthetic clay	DPSCs	The synthetic clay-based hydrogels are a promising biomaterial with no obvious toxic effect on DPSCs and are promising for pulp regeneration	2018	[24]
carboxymethyl cellulose-hydroxyapatite hybrid	DPSCs	carboxymethyl cellulose–hydroxyapatite hybrid hydrogels can be considered promising candidates for pulp complex and periodontal tissue engineering	2015	[25]
Cinnamaldehyde (CMA)	DPSCs	CA crosslinked collagen scaffolds are beneficial to hDPSCs adhesion, proliferation and differentiation	2016	[26]
alginate	DPSCs/HUVECs	Rgd-alginate scaffold produced a microenvironment that significantly enhanced the proliferation of DPSCs/HUVECs combination	2015	[27]

**Table 2 gels-09-00245-t002:** Hydrogels loaded with drugs/bioactive factors used for periodontal anti-inflammatory or regeneration.

Bioactive Factor	Cells	Achievement	Year	Ref.
BMP-2/VEGF	N/A	Both the BMP-2 group and the BMP-2/VEGF group can promote periodontal tissue regeneration	2016	[52]
Bone morphogenetic proteins (BMP)	Periodontal Ligament Stem Cells (PDLSCs.)	Dexgma/gelatin hydrogel scaffolds containing BMP microspheres can promote attachment, proliferation and osteogenic differentiation of PDLSCs.	2007 2019	[53,56]
Matrix metalloproteinase 8 (MMP-8)	N/A	MMP-8 reactive hydrogels can release mmp-8 with antibacterial activity, which has the potential to be used in the treatment of chronic periodontitis and peri-implant inflammation	2019	[54]
Self-assembling peptide (SAP)	PDLSCs	Intermittent systemic parathyroid hormone and locally neutral SAP hydrogels promote periodontal healing	2019	[55]
Insulin-like growth factor-I (IGF-I)	N/A	Locally controlled slow-release IGF-I by adjusting the crosslinking density of hydrogel can promote the regeneration of periodontal membrane and alveolar bone	2006	[57]
Recombinant human beta-nerve growth factor (rh beta-NGF)	N/A	Topical application of rhBMP2 and rh β-ngf improved the quality and quantity of regenerated bone in artificially constructed Beagle Type III bifurcation defects	2010	[58]
Dexamethasone	PDLSCs	Dexamethasone nanocomposite hydrogel effectively reduced periodontitis in the rat model of periodontitis and attenuated inflammatory-induced bone loss.	2022	[59]
Interleukin-1 receptor antagonist (IL-1ra)	mouse mononuclear macrophage leukemia cells (RAW 264.7)	Heat-sensitive hydrogel loaded with il-1ra can effectively inhibit periodontal inflammation and reduce alveolar bone resorption in diabetic periodontitis rats	2022	[60]
Kinase 3 beta inhibitor (BIO)	mouse embryonic osteoblast progenitor cells (MC3T3-E1)	PF127-BIO hydrogel treatment is highly effective in preserving alveolar bone and ligaments and preventing periodontal inflammation in rats	2020	[61]
Minocycline and zinc oxide nanoparticals (ZnO NPs)	gingival cell	Compared to Perio^®^ (2% Minocycline ointment), hydrogels have a significant therapeutic effect and the ability of gum tissue to repair itself	2019	[62]
Chlorhexidine (CHX)	BMSCs	Chx-supported hydrogel has good antibacterial action against Enterococcus faecalis and can promote alveolar bone regeneration	2020	[63]
interleukin (IL)-4/stromal cell-derived factor (SDF)-1α	BMSCs	Simultaneous use of immunomodulators and homing factors in high-hardness hydrogels has been shown to induce stem cell homing, regulate cell differentiation, and induce periodontal tissue regeneration.	2019	[64]
Chlorhexidine	N/A	Hydrogels containing chlorhexidine stop the growth of oral streptococcus and Clostridium albicans	2021	[65]
Naringin	PDLSCs	Naringin—CHC-β-Gp-glycerol colloid hydrogel can inhibit the induction of experimental periodontitis, and has good treatment and inflammatory response	2016	[66]
Metronidazole (MD)	mouse embryonic fibroblast (NIH 3T3)	MD/PAA hydrogel showed good antibacterial activity against Escherichia coli, Staphylococcus aureus and Streptococcus mutans without cytotoxicity	2019	[67]
Ornidazole	PDLSCs	The number of defective new bone and cementum implanted with BMP-7/ORN hydrogel increased significantly	2019	[56]
Triclosan (TCS)/flurbiprofen (FLB) [68]	N/A	Hydrogels containing TCS/PLB have anti-inflammatory and antibacterial effects in rats	2019	[68]
Moxifloxacin hydrochloride (Mox)/clove Essential oil (CEO)	N/A	Hydrogels containing MOX/CEO showed high antibacterial activity against both gram-positive and Gram-negative bacteria	2021	[69]

**Table 3 gels-09-00245-t003:** Application of hydrogel in mineralization of odontogenic stem cells.

Cell	Achievement	Year	Ref.
hDPCs/HUVECs	The differentiation and mineralization ability of odontoblast in the coculture group was significantly enhanced	2020 2015	[15,85]
	The 3D-printed Alg-Gel was more suitable for the growth of hDPSCs, and the scaffold extract could better promote cell proliferation and differentiation	2019	[40]
	RAD/Dentonin has good biocompatibility and can promote the adhesion proliferation, migration, odontogenic differentiation and mineralization of hDPSCs	2021	[28]
	Amylogenic peptide hydrogels promoted the odontogenic differentiation and enhanced the mineralization of hDPSCs	2022	[86]
hDPSCs	Hyaluronic acid hydrogels release NP928, Wnt/β-catenin activity in hDPSCs, and promote restorative dentin formation	2021	[87]
	GelMA promotes osteogenic differentiation of DPSCs and expresses two key matrix proteins, osteopontin and osteocalcin	2021	[88]
	In 3D cell culture system, the characteristics of DN hydrogel promoted the in vitro odontogenic differentiation and mineralization of hDPSCs	2023	[89]
	Most of the DPSCs cultured in PEG-hydrogel maintained circular aggregation and showed the greatest enhancement of tooth-promoting gene expression	2015	[22]
	Higher cell viability, calcium deposition, and alkaline phosphatase activity were observed in GelMA	2022	[90]
hiPSC-MSCs/hDPSCs/hBMSCs	All cells proliferated and differentiated into bone lineages within CPC hydrogel fibers	2016	[91]
hSHED	The use of 3D high-density collagen scaffolds promoted the differentiation and mineralization of SHED bone/odontoblast cells	2013	[92]
hMSCs	The expression levels of RUNX2, COL1AI, SP7 and BGLAP in the hydrogels-induced differentiation in the osteogenic direction were higher than those of MSC in traditional cell culture	2015	[93]
hPDLSCs/hDFCs	RGD additive may promote the application of hydrogel in the mineralized tissue engineering of hPDLSCs and hDFCs	2009	[94]
hPDLSCs	PDLSCs can adhere, survive, migrate, and proliferate on HydroMatrix, which also supports their osteogenic differentiation	2018	[95]

**Table 4 gels-09-00245-t004:** Application of hydrogel in mineralization of enamel.

Hydrogel	Achievement	Year	Ref.
Chitosan/agarose	Chitosan-agarose hydrogels can regenerate layered HAP structures similar to those of natural enamel at the nano to micro scales	2021	[99]
QP5	An amelogenin-derived peptide named QP5 can promote enamel re-mineralization by directing calcium and phosphorus ions provided by bioactive glass (BG)	2022	[100]
Amelogenin-chitosan hydrogel	Amelogenin-chitosan hydrogel can promote significant and lasting enamel repair	2014	[101,102]
	CS-AMEL hydrogel containing MMP-20 significantly increased the modulus and hardness of the repaired enamel (1.8 times and 2.4 times, respectively).	2018	[103]
	CS-AMEL hydrogel can effectively repair erosion and caries under ph cycling conditions	2016	[104]
Agarose hydrogel	Agarose hydrogels can deposit minerals on the enamel surface after demineralization, and the density increases with time	2018	[105]
	The elastic modulus and nano hardness of regenerated enamel prismatic structure are similar to those of natural enamel	2014	[106]
	As a re-mineralized microenvironment, hydrogels initiate the occlusion of dentin tubules and the formation of prism-like tissue of enamel on the dentin surface	2016	[107]
Enamel matrix derivative (EMD)	EMD promotes in vitro biomimetic mineralization and promotes prismatic formation of human enamel after demineralization	2014	[108]
Alginate-carboxymethyl chitosan hydrogels	Oxidized alginate—carboxymethyl chitosan hydrogel can regenerate enamel tissue	2022	[109]
Odontogenic ameloblast-associated protein (ODAM) hydrogels	ODAM promotes HA nucleation in a dose-dependent manner in SBF	2018	[110]
Amelelogenin	The surface microhardness of the re-mineralized enamel was significantly improved by combining the recombinant full-length amelogenin protein with fluoride	2012	[111]
Self-assembled beta-sheet peptide	Self-assembled β-sheet peptide D8 can be used as a template to induce HAP nucleation and promote biomimetic re-mineralization of early caries.	2020	[112]

**Table 5 gels-09-00245-t005:** Application of hydrogel in mineralization of dentin.

Hydrogel	Achievement	Year	Ref.
AMTN gel	AMTN gel can form hydroxyapatite deposits on and within collagen substrates. Coating dentin with rhAMTN promoted the precipitation of surface mineral deposits	2022	[116]
agarose hydrogel	The dentin after demineralization is re-mineralized and the dentin tubules are blocked by growing HA crystals	2017	[114]
	The agarose hydrogel combined with a new electric field-assisted biomimetic mineralization system can re-mineralize the completely demineralized dentine collagen matrix	2015	[117]
Hydroxypropy-lmethylcellulose (HPMC)	Hydroxyl and methoxy groups in HPMC can assist the stability of PAsp-ACP nanoparticles, maintain their biomimetic mineralization activity, and increase the thickness of dentin mineralization (3–4 µm).	2021	[118]
Methacrylic acid functional hydrogel	Gel-MA/NGR1 has a strong ability to promote restorative dentin formation	2021	[119]
Recombinamer-based hydrogels	After 28 days of mineralization, mineral density can reach 1.9 g/cm^3^, which is comparable to that of natural bone and dentin	2015	[120]
GelMA	Loaded dexamethasone-modified hydrogels may have the ability to trigger in situ mineralized tissue regeneration under inflammatory conditions	2021	[121]

## Data Availability

Data sharing not applicable. No new data were created or analyzed in this study. Data sharing is not applicable to this article.

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
