# Peer review of "Research Advances on Hydrogel-Based Materials for Tissue Regeneration and Remineralization in Tooth"

_gels, 2023, doi:10.3390/gels9030245_

Round 1

Author Response

Reviewer #1:

General comment

The manuscript (review) entitled: Research advances on hydrogel-based materials for tissue regeneration and remineralization in tooth, needs some improvements.

Response:We sincerely appreciate your time and effort in reviewing this manuscript. Thank you for your meticulous review for the accuracy and scientificity of the manuscript. Your comments are very valuable and helpful for our revising work. Our point by point responses to your comments are as follows:

  1. Lines 41 to 44a:This part:”kind of high polymer water-soluble polymer material,which has the strecture of”can be removed to better understand what exactly a hydrogel is.

Response: Thank you so much for your careful review.

We have removed “kind of high polymer water-soluble polymer material, which has the strecture of” and re-written the sentence. “Hydrogel is a high polymer crosslinked network with three dimensional stability formed by hydrophilic polymer and provide a porous hydrophilic microenvironment that facilitates the diffusion of oxygen and nutrients.”

(Revision: Line 52 )

  1. line 46:Please be more specific with the expression “hydrogels are cured in a specific environment”and give a reference

Response: We are grateful for your constructive remark.

We have added explanation about “hydrogels are cured in a specific environment” in our manuscript, mainly reffering that hydrogels are sensitive to temperature and UV light, and that hydrogels could be cured under set conditions. “Thermosensitive hydrogels(Shi 2021 ) and photosensitive hydrogels (Cai 2019)form fixed shapes when exposed to specific temperatures or wavelengths of light”

(Revision: Line 58)

  1. line 49:I think that the expression “biotissue”…hydrogel,is a little bit exaggerated and somehow incorrect.Please revise or explain

Response: Thank you for pointing out this problem in manuscript. “Biotissue” is modified to “biotissue-related”, which means that hydrogels could be derived from biological tissue (acellular tissue) or they could serve as natural tissue structure analogues (biomimetic).

(Revision: Line 62)

  1. The abbreviations (as for exmple “hAPCs”from table 1,line 84/85…and so on)should appear explained at their first appearance in manuscript,so that the readers of this journal to be properly informed,especially since it is a “journal on physical(supramolecular)and chemical gel-based materials”for researchers with various interest,not only for medical interest.

Response: We appreciate your kind and constructive comments.

We have had the abbreviations explained at their first appearance in manuscript.

  1. The information from the tablrs are useful but I suggest to use,at the head of the table,the word “achievement” or “performance” in place of the word “conclusion”as this is a word used all the time in the end of an article,to highlight the most important results of the experiments/research papers.

Response: Thank you for your kind suggestion.

We have modified this part by replacing “conclusion”.with “achievement”.

(Revision: Table1-5 )

  1. Subtitle 3.2 is missing(see pages 6 and 7)

Response: We are very sorry for our negligence.

We have added the subtitle.

(Revision: Line 291)

  1. Line 590:Elena is a forename…The surnname is Varoni(see reference 96 )

Response: Thank you for your careful review.

We have made correction according to your comments.

(Revision: Line 652)

  1. I agree with the fact that is more ease to write the name only for the first author fo each reference,but only if it is the rule of the journal.

Response: We gratefully appreciate for your valuable comment.

We have corrected the format of the reference according to the rule of the journal.

Thanks again for your time and efforts in reviewing this manuscript and providing guidance for revising it. Your comments are extremely valuable and helpful during our revising process. Thanks for the guidance and assistance you have dedicated to our work.

1.Shi, J., L. Yu, and J. Ding, PEG-based thermosensitive and biodegradable hydrogels. Acta Biomater, 2021. 128: p. 42-59.

2.Cai, Z., Y. Gan, C. Bao, W. Wu, et al., Photosensitive Hydrogel Creates Favorable Biologic Niches to Promote Spinal Cord Injury Repair. Adv Healthc Mater, 2019. 8(13): p. e1900013.

Reviewer 2 Report

The manuscript entitled „Research advances on hydrogel-based materials for tissue regeneration and remineralization in tooth” shows a comprehensive analysis of the available literature within the field of regeneration of pulp tissue and periodontal tissue. This is a very interesting article. Nevertheless, I cannot accept it in its present form for publication due to some faults/doubts:

1.     In my opinion keywords should not overlap with the title.

2.     You wrote that ‘This paper introduces the latest progress of hydrogel-based materials in pulp and periodontal tissue regeneration and hard tissue mineralization of teeth, and puts forward its application prospect’. Can you specify an exact time period of literature analysis, e.g. last 3-4 years? What tools/websites did you use for this?

3.     The manuscript requires correction due to numerous editorial and linguistic errors.

4.     I noticed that sometimes the Authors provided the full name of the compound/substance with an abbreviation as well as sometimes without an abbreviation and next an unexplained abbreviation appears in the text, e.g. as in the case of cinnamaldehyde, RGD, ZIF-8 etc. Please correct it.

5.     The information in Table 1 is shown in a very chaotic/incoherent way. Namely, the Authors divided the table on Type of hydrogels, Conclusion, Year, Ref. However, in the column marked as ‘Type’, there are not only the name of hydrogels (e.g. Bioactive glass, cinnamaldehyde???)… In turn, in the column marked as ‘Conclusion’, the description is general. Thus, it will be better and clearer if the table will be divided into columns entitled e.g.: (1) Hydrogel, (2) Cells, (3) Results, (4) Ref. . I have the same remark for Tables 2-5. They are unclear.

6.     The Authors should revise the text of section 2.1. Different types of hydrogels for pulp regeneration since there are some substantive mistakes…

7.     In the introduction, there is a lack of a few words about the anatomy of the tooth.

Author Response

The followings are point by point responses to the reviewer’s concerns.

Reviewer #2

General comment

The manuscript (review) entitled: Research advances on hydrogel-based materials for tissue regeneration and remineralization in tooth, needs some improvements.

Responses to REVIEWER 2:

We sincerely appreciate your time and effort in reviewing this manuscript. Thank you for your valuable comments for guiding our revising work. Our point by point responses to your comments are as follows:

  1. In my opinion keywords should not overlap with the title.

Response: Thank you for your comments. 

We used scaffolds instead of hydrogels in the keywords.

(Revision: Line 26 )

  1. You wrote that ‘This paper introduces the latest progress of hydrogel-based materials in pulp and periodontal tissue regeneration and hard tissue mineralization of teeth, and puts forward its application prospect’. Can you specify an exact time period of literature analysis, e.g. last 3-4 years? What tools/websites did you use for this?

Response: Thank you for your patient comments.

We used the pubmed website to search research advances on hydrogel-based materials for tissue re-generation and remineralization from 2014 to 2023. We focused on the research in the last 3-5 years.

(Revision: Line 66 )

  1. The manuscript requires correction due to numerous editorial and linguistic errors.

Response: Thank you for your valuable comments.

We have carefully checked the manuscript, made corrections and arranged language polishing.

  1. I noticed that sometimes the Authors provided the full name of the compound/substance with an abbreviation as well as sometimes without an abbreviation and next an unexplained abbreviation appears in the text, e.g. as in the case of cinnamaldehyde, RGD, ZIF-8 etc. Please correct it.

Response: Thank you for your valuable comments.

We have carefully reviewed and made corrections in this manuscript, listing the full names and corresponding abbreviations when the compound/substance first appear.

  1. Comment:The information in Table 1 is shown in a very chaotic/incoherent way. Namely, the Authors divided the table on Type of hydrogels, Conclusion, Year, Ref. However, in the column marked as ‘Type’, there are not only the name of hydrogels (e.g. Bioactive glass, cinnamaldehyde???)… In turn, in the column marked as ‘Conclusion’, the description is general. Thus, it will be better and clearer if the table will be divided into columns entitled e.g.: (1) Hydrogel, (2) Cells, (3) Results, (4) Ref. . I have the same remark for Tables 2-5. They are unclear.

Response: Thank you for your constructive comments on the Tables. The informations are presented better and more clear after we made adjustment according to your enlightening suggestion.

We have sorted out the contents of Table 1-5 again and carefully modified them.

(Revision: Table 1-5 )

  1. The Authors should revise the text of section 2.1. Different types of hydrogels for pulp regeneration since there are some substantive mistakes…

Response: Thank you so much for your meticulous comment.

We have revised the text of section 2.1 Different types of hydrogels for pulp regeneration to correct the mistakes.

  1. Comment:In the introduction, there is a lack of a few words about the anatomy of the tooth.

Response: Thank you for your valuable suggestion. We have added the anatomy of the tooth section to the introduction. “Tooth is a complex organ with soft and hard tissues of different properties, such as enamel, cementum, dentin, periodontal membrane and pulp. The teeth have a complex root canal network and a sandwich structure of cementum-periodontal membrane-alveolar bone”

(Revision: Line 36)

We are grateful for the precious time you spent on making constructive comments and guidance. Your valuable suggestions have provided so much help during our revising process. 

Reviewer 3 Report

I was pleased to review article ID gels-2261825 entitled “Research advances on hydrogel-based materials for tissue regeneration and remineralization in tooth” for Gels. The review addresses the application of hydrogel-based materials in tissue regeneration and remineralization in teeth. Overall, the article was well written, but some English mistakes and typos are present.

I suggest enriching the following sections in the pulp aspect. As a review, a single or two paragraphs are not enough to address the recent advances in these topics.

2.3. Application of hydrogel microspheres in pulp regeneration 

2.4. Effect of hydrogels on 3D pulp regeneration 

2.5. Regeneration of vascular nerves in dental pulp by hydrogel 

2.6. Antibacterial effect of hydrogels on pulp regeneration 

Suggested articles to add:

Ribeiro, J. S.; Bordini, E. A. F.; Ferreira, J. A.; Mei, L.; Dubey, N.; Fenno, J. C.; Piva, E.; Lund, R. G.; Schwendeman, A.; Bottino, M. C. Injectable MMP-Responsive Nanotube-Modified Gelatin Hydrogel for Dental Infection Ablation. ACS Appl Mater Interfaces 2020, 12 (14), 16006-16017.

Ribeiro, J. S.; Daghrery, A.; Dubey, N.; Li, C.; Mei, L.; Fenno, J. C.; Schwendeman, A.; Aytac, Z.; Bottino, M. C. Hybrid Antimicrobial Hydrogel as Injectable Therapeutics for Oral Infection Ablation. Biomacromolecules 2020, 21 (9), 3945-3956.

The article lacks figures. Figures panels can be created by combining and selecting (with authorization) the best images from the several papers cited to show important findings.

Author Response

Reviewer #3:

General comment

I was pleased to review article ID gels-2261825 entitled “Research advances on hydrogel-based materials for tissue regeneration and remineralization in tooth” for Gels. The review addresses the application of hydrogel-based materials in tissue regeneration and remineralization in teeth. Overall, the article was well written, but some English mistakes and typos are present.

Responses to Reviewer 3:

We sincerely appreciate your time and effort in reviewing this manuscript. Thank you for your constructive comments. They really helped us a lot in revising the manuscript. Our point by point responses to your comments are as follows:

  1. I suggest enriching the following sections in the pulp aspect. As a review, a single or two paragraphs are not enough to address the recent advances in these topics.

Response: Thank you for pointing out this problem in our manuscript. We have enriched the manuscript by adding content concerning microspheres/3D/vascular nerves/antibacterial as you suggested. Plus, We have supplemented the articles which really helped us elevate the manuscript’s quality. Thank you again for your enlightening guidance.

(Revision: Line 148,Line 161,Line 184,Line 217,Line 243)

  1. The article lacks figures. Figures panels can be created by combining and selecting (with authorization) the best images from the several papers cited to show important findings.

Response: Thank you for your constructive suggestions. Indeed, figures are there to render better illustration for the composer and better understanding for the audience. We have added new figures into the manuscript.

Figure 2. Application of hydrogel microspheres in pulp regeneration

Figure 4. Application of hydrogel in hard tissue regeneration of teeth.

(Revision: Figure 2, Figure 4)

  1. Overall, the article was well written, but some English mistakes and typos are present.

Response: Thank you for your kind suggestion. We have carefully reviewed the manuscript and correct the language mistakes and typos.

We sincerely appreciate your valuable comments which are really helpful during our revising process. We have made revisions according to those comments. Thanks again for your time and efforts in reviewing this manuscript and providing guidance for revising it.

Round 2

Reviewer 1 Report

Interesting and useful article in revised form.

Reviewer 2 Report

Dear Authors,

Thank you for responding to my comments and making corrections. Now I can recommend your manuscript for publication.